# Immunologic Dysregulation and Hypercoagulability as a Pathophysiologic Background in COVID-19 Infection and the Immunomodulating Role of Colchicine

**DOI:** 10.3390/jcm10215128

**Published:** 2021-10-31

**Authors:** Dimitrios A. Vrachatis, Konstantinos A. Papathanasiou, Sotiria G. Giotaki, Konstantinos Raisakis, Charalampos Kossyvakis, Andreas Kaoukis, Fotis Kolokathis, Gerasimos Deftereos, Konstantinos E. Iliodromitis, Dimitrios Avramides, Harilaos Bogossian, Gerasimos Siasos, George Giannopoulos, Bernhard Reimers, Alexandra Lansky, Jean-Claude Tardif, Spyridon Deftereos

**Affiliations:** 1Medical School, National and Kapodistrian University of Athens, 11527 Athens, Greece; dvrachatis@gmail.com (D.A.V.); kpapathanasiou91@gmail.com (K.A.P.); sotiria.giotaki@yahoo.com (S.G.G.); fotisks@hotmail.com (F.K.); ger_sias@hotmail.com (G.S.); 2Department of Cardiology, “G. Gennimatas” General Hospital of Athens, 11527 Athens, Greece; kraisakis@yahoo.co.uk (K.R.); ckossyvakis@gmail.com (C.K.); andreaskaoukis@yahoo.gr (A.K.); gerasimosd@gmail.com (G.D.); d_avramides@yahoo.com (D.A.); 3Evangelisches Krankenhaus Hagen-Haspe, Clinic for Cardiology and Electrophysiology, 58135 Hagen, Germany; konstantinos.iliodromitis@gmail.com (K.E.I.); bogossian@evk-haspe.de (H.B.); 4Department of Cardiology, University of Witten/Herdecke, 58455 Witten, Germany; 5Medical School, Artistotle University of Thessaloniki, 54124 Thessaloniki, Greece; ggiann@med.uoa.gr; 6Humanitas Research Hospital IRCCS, Rozzano, 20089 Milan, Italy; bernhard.reimers@humanitas.it; 7Section of Cardiovascular Medicine, Department of Internal Medicine, Yale School of Medicine, New Haven, CT 06510, USA; alexandra.lansky@yale.edu; 8Montreal Heart Institute, Université de Montréal, Montreal, QC H1T 1C8, Canada; Jean-Claude.Tardif@icm-mhi.org

**Keywords:** COVID-19, cytokine release syndrome, hypercoagulability, colchicine, cardioprotection

## Abstract

In 2020, SARS-COV-2 put health systems under unprecedented resource and manpower pressure leading to significant number of deaths. Expectedly, researchers sought to shed light on the pathophysiologic background of this novel disease (COVID-19) as well as to facilitate the design of effective therapeutic modalities. Indeed, early enough the pivotal role of inflammatory and thrombotic pathways in SARS-COV-2 infection has been illustrated. The purpose of this article is to briefly present the epidemiologic and clinical features of COVID-19, analyze the pathophysiologic importance of immunologic dysregulation and hypercoagulability in developing disease complications and finally to present an up-to-date systematic review of colchicine’s immunomodulating capacity in view of hindering coronavirus complications.

## 1. Introduction: Coronavirus Disease 19 (COVID-19) Pandemic

On December 2019 Wuhan, China came to public attention due to emerging cases of fever of unknown origin and on 7 January 2020, WHO revealed the isolation of a new coronavirus strain (SARS-COV-2). At the end of the same month, China declared 7000 cases of COVID-19 infection and the virus had already spread to eighteen countries worldwide, hence establishing a new pandemic [1]. By early September 2021, more than 200 million people have been infected and the death toll has already surpassed 4 million [2].

## 2. COVID-19 Clinical and Pathophysiological Aspects

### 2.1. COVID-19 Infection: Clinical Course

Severe Acute Respiratory Syndrome Coronavirus 2 (SARS-COV-2) belongs to the coronaviridae family of viruses. All four subfamilies (α,β,γ,δ) of coronaviridae viruses affect humans or animals with respiratory tract, gastrointestinal tract and central nervous system infections [1]. Humans are susceptible only to subfamilies alpha and beta, with SARS-CoV and Middle-East Respiratory Syndrome (MERS)-CoV being examples of beta coronaviruses. SARS-COV-2 belongs to the beta subfamily and is a single RNA virus with an external envelope [1]. Both viral and host genetic variations are critical in understanding transmissibility, pathogenicity and severity of COVID-19 infection [3,4]. Taking advantage of the molecular mimicry between glycoprotein S [5,6] and angiotensin converting enzyme-2 (ACE-2) membrane receptor, SARS-COV-2 invades host cells and affects many organs (heart, endothelium, liver, lungs, gastrointestinal tract, kidney, bladder) [7]. Contaminated droplets represent the mainstay of viral transmission between humans [8]. The viral inoculation varies with a mean time to symptom development of five days. Comorbidities, obesity and age are implicated in disease severity, complication development and bacterial superinfection [9,10]. Approximately 80% of infected patients develop mild flu-like or diarrhea manifestations, 15% present with pneumonia and hypoxemia, and the remaining 5% suffer multi-organ failure with high mortality (up to 50%) [10]. The imaging hallmark is widespread and peripherally located ground glass opacities in the lungs, detected in 75% of cases on computed tomography imaging [1]. The second immunologic phase of COVID-19 infection commonly occurs during the second week after symptom onset, and patients may suffer a wide variety of complications such as acute respiratory distress syndrome, acute kidney injury, myocardial injury, sepsis, rhabdomyolysis, liver failure and venous thromboembolic disease [11]. Lymphocytopenia, prothrombin time prolongation and elevated LDH are typical laboratory alterations. Marked cytokine release (IL-1B, IL-1RA, IL-6, IL-7, IL-8, IL-2R, TNF-a) is also detected in serum [12,13]. The diagnosis is confirmed by molecular testing of nasopharyngeal swab or sputum for SARS-COV-2 RNA. During the first year of the pandemic (2020), preventive measures such isolation of infected cases, social distancing and hand hygiene were adopted. Vaccines became publicly available late in 2020 and additional efforts are underway [14]. 

The main therapeutic modalities tested against SARS-COV-2 fall in the following categories (Table 1): (1) active or passive immunization via vaccination and convalescent plasma, or immunoglobulins, respectively; (2) antiviral drugs (fusion inhibitors, protease inhibitors, RNA polymerase inhibitors, endosomal acidification inhibitors); (3) symptom-based device therapies such as ECMO (extracorporeal membrane oxygenation), ALS (artificial liver system), cytokine filters; (4) immunomodulation: colchicine, azithromycin, corticosteroids, interferons a or b, thalidomide, JAK inhibitors, IL-1 or IL-6 inhibitors, BTK inhibitors and cell therapies [15,16,17,18,19].

### 2.2. COVID-19 Pathophysiology: Immunologic Dysregulation and Hypercoagulability 

The infection has been suggested to have an initial flu-like phase followed by a second immunologic stage [13]. The underlying pathophysiologic component of the latter features a marked cytokine release, thus researchers named it cytokine storm or cytokine release syndrome (alternatively known as macrophage activation syndrome or secondary hemophagocytic lymphohistiocytosis) [20]. This immunologic dysregulation differentiates SARS-COV-2 from previous coronaviruses and influenza, and is also critically involved in complication development and increased mortality [20]. Viral infections, in general, trigger an immunologic reaction that culminates in inflammasome activation and IL1-b and IL-18 production [21,22]. SARS-COV-2 leads to hyperactivation of innate and adaptive immune reactions [19]. After invading lung alveolar cells, the virus takes advantage of the host enzymatic machinery to replicate and ultimately brings about cell lysis and neighbouring cell invasion. Antigen-presenting cells (dendritic cells and macrophages) activate CD4+ lymphocytes, which in turn mobilize CD8+ lymphocytes and then neutrophils and macrophages. CD8+ lymphocytes destroy infected alveolar cells, while neutrophil and macrophage activation leads to the systematic inflammatory reaction syndrome (SIRs) and cytokine release syndrome [23]. Complement component C5a, acting as chemoattractant of neutrophils, potentiates the development of a deleterious hypercytokinaemia [24]. T-helper 17 polarization, occurring after neutrophil hyperactivation, seems to hold a central role in triggering a vicious cycle of inflammation and fibrosis [25]. The prevailing mechanisms employed to explain the immunologic dysregulation apparent in COVID-19 infection are (a) immune system escape [26]; (b) reduced INF-γ release [27,28]; (c) NETosis: a complex apoptotic mechanism which leads to extracellular neutrophilic traps [29,30]; (d) pyroptosis: another apoptotic pattern induced by rapid viral replication and cytokine overexcretion [31]. The final immunologic outcome is a state of reduced lymphocyte activity counterbalanced by macrophage hyperactivation. [20]. Severe cases feature a hypercoagulable state that resembles the prothrombotic manifestations of disseminated intravascular coagulation [32]. 

At least four contributory mechanisms to hypercoagulability are under investigation (Figure 1): autoimmunity and anti-phospholipid antibodies [33,34]; cytokine storm-mediated thrombin overproduction [35]; increased tissue factor and plasminogen activator inhibitor-1 levels due to ARDS-related tissue damage [36]; and ACE2 receptor-mediated endothelial dysfunction and “endothelitis” [37]. The endothelium is central to coagulation physiology, and its dysfunction explains at least partly the clinical complications (venous thromboembolic disease) and the laboratory hallmarks (elevated d-dimer, prothrombin time prolongation) of severe COVID-19 infection cases [12,32]. Of note, inflammation and coagulation share molecular pathways involving cytokines such as IL-8, IL-6, IL-1b [20] and complement components C3 and C5 [24,38]. Immune complexes containing antibodies against SARS-COV2 spike protein are also implicated in COVID-19 hypercoagulability through enhanced of FcγRIIA signaling [39,40]. Besides an hypercoagulative state, the COVID-19-related hyperinflammatory status may culminate in lung pneumonitis and/or fibrosis and/or vasculitis-like coronary and skin lesions [41].

## 3. Colchicine and COVID-19 Infection

### 3.1. Brief History, Pharmacology and COVID-19 Hypothesis

According to German archeologist Ebers, the first clinical application of colchicine dates back to 16 century BC, when Egyptians used the herb Colchicum autumnale for the treatment of swelling [42].

In 1987, Rodriguez de la Serna et al. rekindled physicians interest in studying colchicine, since they reported its beneficial role in managing recurrent pericarditis [43]. Katsilambros examined the antiviral properties of colchicine in a variety of infections (herpes zoster virus, mumps, echovirus and coxsackievirus among others) since 1955 [44], yet a recent systematic review suggested that the clinical value of using colchicine for infectious disease management is still inconclusive [45].

Through the centuries colchicine was proven effective in alleviating many inflammatory diseases such as gout, pericarditis, familial Mediterranean fever, Adamantiades-Behcet’s disease, post-operative pericarditis and a variety of dermatologic conditions [14,46,47,48,49]. SARS-CoV-2-induced hypercytokinaemia might also be positively affected by colchicine ’s anti-inflammatory properties [50]. 

Colchicine is a lipophilic alkaloid and it is metabolized in the liver and primarily excreted via the gastrointestinal tract (80%) [51]. To minimize side effects and drug-drug interactions, physicians should meticulously examine co-administration of medications that are substrates of either CYP3A4 enzyme or P-glycoprotein transporter. Colchicine is neither an inhibitor nor an inducer of these two metabolic pathways, yet substances strongly affecting those two enzymes might increase colchicine plasma concentration to toxic levels [48]. Diarrhea, nausea and vomiting are common side effects after colchicine administration, but most cases are mild and manageable, since it is a dose-related phenomenon. Infrequently, transient myelosuppression or myopathy may occur in the setting of severe kidney or severe liver disease, particularly if the contra-indication related to the use of concomitant medications like clarithromycin is not respected (azithromycin can be used) [52]. Two recent meta-analyses are reassuring and suggest the relative safety of colchicine administration [53,54]. In addition, the two large COLCOT and LoDoCo2 trials, which included a total of more than 10,000 patients with coronary artery disease followed for approximately 2 years, have also shown the safety of low-dose colchicine [55,56].

Colchicine seems to possess pleiotropic mechanisms of action [42,51,52,57,58,59] such as: (1) inhibition of neutrophil chemotaxis, extravasation and degranulation via tubulin depolymerization, (2) reduced IL-1β and IL-18 secretion through NLRP3 (NOD-, LRR- and pyrin domain-containing protein) inflammasome inhibition in neutrophils and macrophages, (3) inhibition of platelet-neutrophil and platelet-monocyte aggregates (notably, platelet to platelet interactions are spared and normal coagulation mechanisms remain intact), (4) maturation of dendritic antigen-presenting cells, (5) anti-fibrotic effects (colchicine induces Bcl-2 gene expression and down-regulates ROCK pathway as well as transforming growth factor-β1), (6) endothelial protection via reduced Lp-PLA2 activity and enhanced nitric oxide levels [60], (7) T-cell inhibition via downregulation of L-selectin [61], (8) anti-vascular endothelial growth factor effect [62,63], (9) calcium homeostasis regulation in cardiomyocytes [64], and (10) suppression of NETs [65,66]. Regarding COVID-19 infection, inhibition of clathrin-mediated endocytocis (at least partly regulated via tubulin polymerization) may be involved in its potentially favorable effects [67,68]. Furthermore, the protease furin belonging to the PCSK (pro-protein convertase subtilisin/kexin) family activates the spike protein of SARS-COV-2 and is implicated in endothelial tropism of the virus [69]. Interestingly, colchicine has been suspected to inhibit the enzymatic activity of furin [69]. Aiming to repurpose existing drugs against SARS-COV-2, Peele et al. carried out an extensive molecular dynamics analysis and found that colchicine is a potent inhibitor of viral assembly by exerting high binding affinity with its membrane (M) protein [70]. The role of tissue factor overexpression is increasingly recognized in COVID-19-related coagulopathy, and the former might be affected by colchicine [71]. Lastly, colchicine reduced acute lung injury in a rat model of acute respiratory distress syndrome [72].

### 3.2. Clinical Results with Colchicine in COVID-19 Patients

The aforementioned multifaceted actions of colchicine, affecting critical stages of inflammation (innate, adaptive immunity, angiogenesis, fibrosis) as well as cardiomyocyte and endothelial functions, encouraged many investigators to put colchicine to the test against COVID-19 infection (see also Figure 2).

A systematic PubMed search was carried out (up to 2 September 2021) using the following query ‘’colchicine’’ AND ‘’COVID’’ OR ‘’COVID-19’’ OR ‘’SARS-COV 2’’ OR ‘’betacoronavirus’’. References cited in the articles initially identified by this query were reviewed in order to identify any supplemental studies (“snowball procedure”). Out of 172 results, 27 studies reported outcomes regarding colchicine utilization against SARS-COV-2 and are discussed in the following sections (Figure 3).

#### 3.2.1. Studies Evaluating Colchicine as Prevention against COVID-19

As could be predicted by known pathophysiology and its properties, studies that evaluated colchicine against the risk of infection with SARS-COV-2 showed no protective effect. Bourguiba et al. conducted a retrospective survey of FMF (Familial Mediterranean Fever) patients on chronic colchicine treatment (1 mg daily) in France from March to May 2020. They identified 27 infected patients and found neither increased risk for severe infection due to FMF history nor protection after chronic colchicine administration [73]. The small number of patients and the observational study design with no controls are the major limitations of this study. Similarly, Güven et al. retrospectively examined data from 34 FMF patients diagnosed with SARS-COV-2 between March and December 2020 in Turkey and their findings were in accordance with the above-mentioned study [74]. In addition, Gendelman et al. retrospectively investigated 14,520 subjects from the Maccabi Health Services database in Israel (from 23 February 2020 to 31 March 2020) and found no difference in the rate of COVID-19 infection between colchicine users and non-users [75].

The protective role of colchicine against COVID-19-related hospital admissions was also investigated in a retrospective observational study in Spain from March 2020 to May 2020. After evaluating 9379 rheumatologic patients on colchicine (71.6% female, median age 61.1 years, 8% diabetes), Madrid-García et al. found no statistically significant change in hospitalization rates (2.96% in colchicine users vs. 1.34% in controls) [76].

A Turkish cohort of 404 pediatric patients suffering from chronic inflammatory disease was investigated in a retrospective manner between 11 March and 15 May 2020. 90% of the study population had a diagnosis of FMF and 93% were on colchicine. Among 376 colchicine users, 6 patients contracted COVID-19 infection and only one of them required hospitalization [77]. The retrospective design, short follow-up and lack of control group are the major limitations of this analysis. A cross-sectional study conducted from June 2020 to February 2021 evaluated the features of 822 patients with FMF and suggested that colchicine non-compliance was not associated with COVID-19 hospitalization. Of note, this Turkish cohort investigated a relatively young population without major comorbidities (mean age 35.6 years, female gender 62.9%, cardiovascular disease 0.2%, and diabetes mellitus 1.2%) [78].

Finally, a small (*n* = 36) retrospective case-control study from Jerusalem revealed that neither FMF history nor colchicine administration at baseline affected COVID-19-related length of hospitalization, mortality, oxygen support needs and mechanical ventilation [79].

#### 3.2.2. Colchicine for the Treatment or Prevention of COVID-19 Complications

Early in the COVID-19 pandemic, colchicine has been proposed as a potential means to alleviate the cytokine storm and hence prevent COVID-19 complications [80,81]. Up to April 2020, only case reports and small case series had been published providing a signal of benefit [82,83,84,85,86]. In the open-label, randomized GRECCO-19 study, Deftereos et al. observed longer time to clinical deterioration of COVID-19 with colchicine compared to the control group [87]. In particular, only 1.8% of patients taking colchicine suffered clinical deterioration as compared to 14.0% of the patients on conventional treatment. Interestingly, the secondary endpoint of peak d-dimer concentration was reduced by colchicine [88]. This observation should be further investigated in view of the hypercoagulable state associated with COVID-19 infection and the prognostic value of d-dimer level in this setting [89].

The results of the double-blind, randomized controlled COLCORONA trial then became available. COLCORONA focused on 4488 community-treated patients who were randomly assigned to receive colchicine (0.5 mg twice daily for 3 days and once daily for the following 27 days) or placebo for 28 days [90]. The effect of colchicine compared to placebo on the primary endpoint (death or hospitalization) was not statistically significant for the entire population (odds ratio 0.79, 95% CI 0.61–1.03, *p* = 0.081). However, in a predefined analysis of patients with PCR-proven diagnosis (4159 patients), the benefit of colchicine on the primary endpoint was statistically significant (4.6% vs. 6.0%, respectively; odds ratio 0.75, 95% CI 0.57–0.99, *p* = 0.042). This effect was primarily driven by hospitalizations rather than deaths. Although not a pre-specified outcome, all-cause hospitalizations were lowered by colchicine administration and reached statistical significance (4.9% vs. 6.3%; *p* = 0.04). The number needed to treat with colchicine in order to prevent a death or hospital admission was 25 and 29 in those with cardiovascular disease and diabetes mellitus, respectively. 

Recovery is an ongoing trial that investigated, among its various arms, colchicine versus standard of care therapy (94% of patients were on systemic corticosteroid treatment) and found no additional benefit on 28-day mortality (risk ratio 1.02; 95% CI 0.94–1.11; *p* = 0.63). Of note, a high mortality rate was observed in both treatment arms (21%), suggesting that this hospitalized population suffered from severe COVID-19 [91]. One study limitation is that colchicine and systemic steroids are rarely used clinically in combination for any inflammatory disease.

The ECLA PHRI COLCOVID results were announced at the European Society of Cardiology Virtual Congress in August 2021 [92]. This open-label study randomized 1277 hospitalized adults (mean age 62 years, 64% male) to standard care with or without colchicine (loading dose of 2 mg on day 1 and 0.5 mg twice daily for 2 weeks). At 28 days of follow-up, the effect of colchicine on the co-primary endpoint of death or mechanical ventilation did not reach statistical significance (HR: 0.83; 95% CI: 0.67–1.02; *p* = 0.08). Of note, the occurrence of the composite endpoint of new intubation or death due to respiratory failure was reduced in the colchicine group compared to the control group (HR 0.79%; 95% CI 0.63–0.99; *p* < 0.05) [92]. These results are concordant with those of other studies in hospitalized patients with SARS-COV-2 that have shown benefits of colchicine on ordinal scale clinical deterioration [92] and need for supplemental oxygen [92] (see Table 2). 

Finally, Karatza et al. conducted a pharmacokinetic simulation analysis showing that 0.5 mg of colchicine twice daily achieves therapeutic plasma levels with minimal toxicity [101].

#### 3.2.3. Meta-Analyses

A series of six studies of patients hospitalized for COVID-19 were meta-analyzed by Vrachatis et al. and a mortality risk reduction was found in those treated with colchicine (OR 0.35; 95% CI 0.24–0.52) [102]. Updated meta-analyses confirmed these findings. Salah et al. conducted a meta-analysis of 8 studies (5259 patients) and found that colchicine reduced mortality (3.2% vs. 8.3%; RR: 0.62; 95% CI: 0.48–0.81) [103]. Similar results were reported by Hariyanto et al. [104]. Chiu et al. included six studies (3 observational studies and 3 randomized trials) having included 5033 patients and reported mortality risk reduction with colchicine administration (OR 0.36; 95% CI: 0.17–0.76), but this survival benefit was no longer significant when analyzing only randomized trial data (OR 0.49; 95% CI: 0.20–1.24) [105]. More recent meta-analyses of COVID-19 studies have also reported benefits of colchicine administration on mortality [106,107,108,109]. Of note, Nawangsih et al. demonstrated that colchicine’s survival benefit is more pronounced in younger adults yet, remains statistically important irrespective of diabetes and coronary artery disease history [108]. The incorporation of RECOVERY trial findings in a meta-analysis conducted by Lien et al. did not alter the afore-mentioned survival benefit among colchicine treated patients [109] (see also Table 3).

Recently, the Italian Society of Anti-Infective Therapy and the Italian Society of Pulmonology guidelines highlighted that colchicine may be considered in outpatients with COVID-19, if other randomized trials replicate COLCORONA’s favorable findings [110]. 

Finally, a limitation of our study is that we opted for data published in MEDLINE instead of unpublished papers or studies published in other databases. Another limitation is the fact that the presented data predate the new era of vaccination against SARS-COV2.

## 4. Conclusions

SARS-COV-2 not infrequently triggers a massive release of cytokines, capable of causing severe lung injury and multi-organ dysfunction and establishing a detrimental prothrombotic state. Colchicine is an old, safe, inexpensive but powerful anti-inflammatory medication with favorable pleiotropic and cardioprotective effects. Currently available data have shown colchicine’s potential in SARS-COV2 infected patients. Further research will clarify its role in the ongoing COVID-19 pandemic.

## Figures and Tables

**Figure 1 jcm-10-05128-f001:**
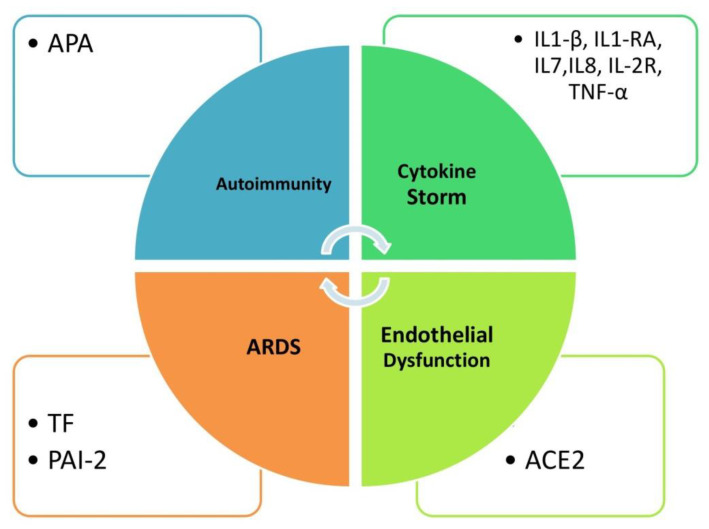
Contributory mechanisms of hypercoagulability in COVID 19 infection are autoimmunity and anti-phospholipid antibodies, cytokine storm-mediated thrombin overproduction, increased tissue factor and plasminogen activator inhibitor-2 levels due to ARDS-related tissue damage and ACE2 receptor-mediated endothelial dysfunction. APA anti-phospholipid antibodies, IL1RA interleukin 1 receptor agonist, IL-2R interleukin 2 receptor, ARDS acute respiratory distress syndrome, TNF tissue necrosis factor, ACE angiotensin converting enzyme, TF tissue factor, PAI-2 plasminogen activator inhibitor-2.

**Figure 2 jcm-10-05128-f002:**
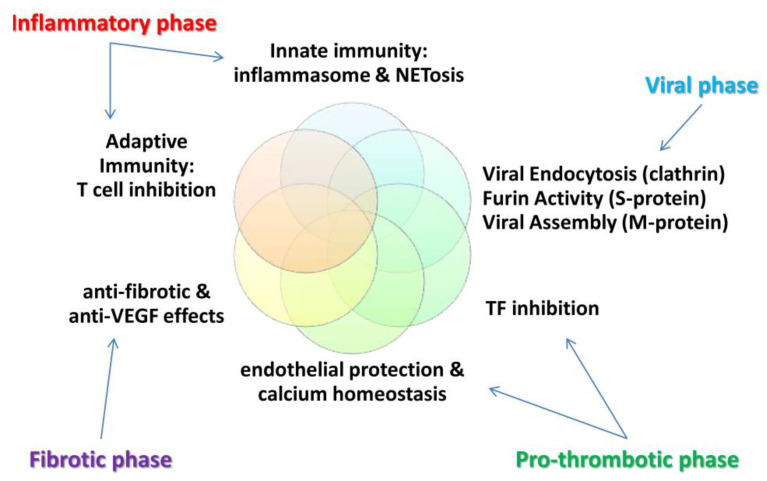
Colchicine’s mechanisms of actions generate the multifaceted hypothesis against COVID-19 infection.

**Figure 3 jcm-10-05128-f003:**
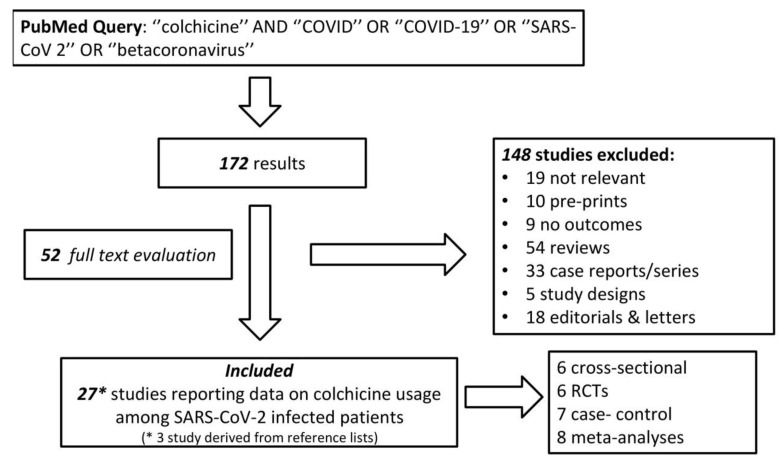
PubMed Query Flowchart.

**Table 1 jcm-10-05128-t001:** Therapeutic modalities against COVID-19 infection.

Therapeutic Strategy	Agent/Approach Tested
Immunization	active immunity: vaccines passive immunity: plasma, immunoglobulins
Antiviral Agents	fusion inhibitors RNA polymerase inhibitors protease inhibitors endosome acidification inhibitors
Supportive Care	ECMO artificial liver system (ALS) cytokine filters thromboprophylaxis
Immunomodulation	Interferons Steroids Colchicine Macrolides JAK inhibitors BTK inhibitors IL-6 inhibitors IL-1 inhibitors Anti-GM-CSF Abs Thalidomide Cell therapies

ECMO, extracorporeal membrane oxygenation; JAK, Janus kinase; BTK, bruton tyrosine kinase; IL, interleukin; anti-GM-CSF Abs, antibodies against granulocyte-macrophage colony stimulating factor.

**Table 2 jcm-10-05128-t002:** Main characteristics and outcomes of completed studies investigating the impact of colchicine in SARS-COV-2 infected patients.

Author Country Year	Design	*n*	Age	♀ (%)	BMI (kg/m^2^)	DM (%)	Colchicine Dosage	SoC	Findings	Outcome
Randomized Controlled Trials										
Deftereos SG [87] Greece April 2020	prospective, open-label, RCT	105	64	41.9	27.5	10.5	L:2 mg M: 0.5 mg BD up to 21 days	HCQ ± AZM ± LPV/r ± tocilizumab	WHO-OSCD: 1.8% vs. 14% OR 0.11; 95% CI: 0.01–0.96; *p*: 0.02	Positive
Tardif JC et al. [90] Multinational March 2020–January 2021	double-blinded RCT non-hospitalized patients	4488	54	53.9	30	19.9	0.5 mg BD 3d 0.5 mg OD 27d	HCQ OACs Anti-platelets	Composite of death or hospitalization 4.7% vs. 5.8% OR:0.79; 95% CI: 0.61–1.03; *p*: 0.081 Secondary endpoint, Mechanical ventilation OR: 0.53; 95% CI: 0.25–1.09 secondary analysis PCR-confirmed COVID-19 Composite of death or hospitalization 4.6% vs. 6% OR: 0.75; 95% CI: 0.57–0.99; *p*: 0.042 Any hospital admission (not pre-specified) OR: 0.76; 95% CI: 0.58–0.99; *p*: 0.04	Neutral
Lopes M. et al. [93] Brazil April–August 2020	double blinded RCT	72	55	54	31.6	39	0.5 mg q 8 h 5 days 0.5 mg BD 5 days	AZM 500mg OD up to 7 days HCQ 400 mg BD daily for 2 days, then 400 mg OD up to 8 days	Need for supplemental oxygen 4.0 days vs. 6.5 days; *p* < 0.001 Time of hospitalization 7.0 days vs. 9.0 days; *p*: 0.003	Positive
Mareev VY et al. [94] (COLORIT study) Russia	prospective comparative RCT	43	61	30	30.4	11.6	1 mg/day for 3 days 0.5 mg/day for 14 days	NA	Primary endpoint SHOCS-COVID score change day 12 8 to 2 vs. 7 to 7 (*p*: 0.002) Secondary outcomes Hospital stay: 13 vs. 17.5; *p*: 0.079 Any oxygen support: decreased from 50% to 9.5%; *p*: 0.011 Deaths: 0 vs. 2; *p*: 0.467	Positive
Horby PW et al. [91] Pre-print RECOVERY NCT04381936 Multinational November 2020–March 2021	open-label, RCT	11340	63.4	30.5	NA	25.5	L: 1 mg + 0.5 mg 12 h later M: 0.5 mg BD up to 10 days	corticosteroids remdesivir tocilizumab convalescent plasma baricitinib aspirin	Primary outcome 28-day all-cause mortality RR: 1.01; 95% CI: 0.93–1.0; *p*: 0.77 Secondary outcomes time to discharge 10 vs. 10 d invasive mechanical ventilation 11 vs. 11%	Neutral
Observational Studies										
Brunetti L et al. [95] USA March–May 2020	propensity matched retrospective observational cohort	66	62.9	34.8	30.7	21.2	L: 1.2 mg M: 0.6 mg BD	AZM, HCQ ± remdesivir or tocilizumab	28-days mortality OR:0.20; 95% Cl:0.05–0.80; *p*: 0.023 WHO-OSCI days 14 and 28 57.6% vs. 51.5%; *p*: 0.621 Not requiring supplemental oxygen on days 14 and 28 54.5% vs. 54.5%, *p*: 1.0 Hospital discharge by day 28 OR: 5.0; 95% CI: 1.25–20.1; *p*: 0.023	Positive
Scarsi M et al. [96] Italy March–April 2020	retrospective, case-control observational study	262	70	36.5	NA	NA	M: 1 mg/day	HCQ ± dexamethasone ± LPV/r	21 days survival rate: 84.2% vs. 63.6%, *p*: 0.001 adj HR: 0.151; 95% CI: 0.062–0.368; *p* < 0.0001	Positive
Sandhu T et al. [97] USA March–May 2020	prospective comparative cohort study	112	67	43	27.5	42	0.6 mg BD 3 days 0.6 mg OD for 9 days	HCQ, steroids, oseltmamivir Excluded if on: lamivudine dolutegravir, tocilizumab convalescent plasma	Mortality: 49.1% vs. 72.9%; *p*: 0.002 Intubations: 52.8% vs. 73.6%; *p*: 0.006 Discharge rate: 50.9% vs. 27.1%; *p*: 0.002	Positive
Kevorkian JP et al. [98] (COCAA-COLA study) France January–November 2020	observational cohort study	68	66	22	27	44	Prednisone 1 mg/kg/day Furosemide 80 mg/day Salicylate 75 mg/day Colchicine L: 1.5 mg M: 0.5 mg q 8 h Rivaroxaban or Apixaban	dexamethasone (6 mg OD for up to 10 days) LMWH	Primary composite endpoint OR: 0.097; 95% CI: 0.001–0.48; *p*: 0.0009 requirement of HFOT: 13% vs. 4%; *p*: 0.38 non-invasive mechanical ventilation: 13% vs. 0%; *p*: 0.07 invasive mechanical ventilation: 15% vs. 4%; *p*: 0.21 28-day mortality: 5% vs. 0%; *p*: 0.5	Positive
Manenti L et al. [99] Italy February–April 2020	retrospective cohort study age & sex matched	141	61.5	29	27.5	17	M: 1 mg/day up to 21 days	ABX, antivirals, HCQ, i.v steroids, tocilizumab	21 days mortality adj HR: 0.24; 95% CI: 0.09–0.67; *p*: 0.006 WHO-OSCI: adj relative improvement rate 1.80 95% CI: 1.00–3.22; *p*: 0.048	Positive
García-Posada M et al. [100] Colombia May–August 2020	descriptive observational study	209	60	39	NA	25.3	L:2 mg M: 0.5 mg BD up to 20 days	varying combinations of ABX corticosteroids LMWH or tocilizumab	All-cause mortality: (combination of ABX, LMWH, colchicine, corticosteroids) OR: 0.26; 95% CI: 0.08−0.71; *p* < 0.05	positive

Abbreviations; *n*: number of study participants, female symbol, DM: diabetes mellitus, SoC: standard of care, L: loading dose, M: maintenance dose, ABX: antibiotics, BD: twice daily, OD: once daily, RCT: randomized clinical trial, WHO-OSCI: ordinal scale clinical improvement, WHO-OSCD: ordinal scale clinical deterioration, LPV/r: Lopinavir /ritonavir, AZM: azithromycin, HCQ: hydroxychloroquine, OACs: oral anticoagulants, LMWH: low molecular weight heparin, HFOT: high flow oxygen therapy, SHOCS-COVID score: assessment of the patient’s clinical condition, computed tomography (CT) score of the lung tissue damage, CRP changes, D-dimer changes.

**Table 3 jcm-10-05128-t003:** 8 meta-analyses reporting effect of colchicine administration in COVID-19 mortality. (Abbreviations: OR odds ratio, RR relative risk, CI confidence interval, *n*/R not reported, *n* population, RCTs randomized clinical trials).

Meta-Analysis	Journal	*n*	Studies Included	Mortality Effect
Vrachatis et al., 2021 [102]	Hell J Cardiol	881	6 studies 3 cohorts 2 RCTs 1 case-control	OR 0.35 (95% CI: 0.24–0.52; *p* < 0.05)
Salah et al., 2021 [103]	Am J Cardiol	5259	8 studies 4 cohorts 3 RCTs 1 case-control	RR 0.62 (95% CI: 0.48–0.81; *p* = 0.0005)
Hariyanto et al., 2021 [104]	Clin Exp Pharmacol Physiol	5778	8 studies 4 cohorts 3 RCTs 1 case-control	OR 0.43 (95% CI: 0.32–0.58; *p* = *n*/R)
Chiu et al., 2021 [105]	medRxiv (pre-print)	5033	6 studies 3 cohorts 3 RCTs	OR 0.36 (95% CI: 0.17–0.76; *p* = *n*/R)
Golpour et al., 2021 [106]	Int J Immunopathol Pharmacol	5678	10 studies 5 cohorts 4 RCTs 1 case-control	RR 0.365 (95% CI: 0.555–0.748; *p* < 0.05)
Elshafei et al., 2021 [107]	Eur J Clin Invest	5522	9 studies 4 cohorts 4 RCTs 1 case-control	OR 0.35 (95% CI: 0.25–0.48; *p* = *n*/R)
Nawangsih et al., 2021 [108]	Int Immunopharmacol	5530	8 studies 5 cohorts 3 RCTs	OR 0.47 (95% CI: 0.31–0.72; *p* = 0.001)
Lien et al., 2021 [109]	Life (Basel)	17,205	11 studies 7 cohorts 4 RCTs	OR 0.57 (95% CI: 0.38–0.87; *p* < 0.01)

## Data Availability

Data available on request.

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
