# Peer review of "Immunologic Dysregulation and Hypercoagulability as a Pathophysiologic Background in COVID-19 Infection and the Immunomodulating Role of Colchicine"

_jcm, 2021, doi:10.3390/jcm10215128_

Round 1

Reviewer 1 Report

Table 1: graphic may be imporved

Fig. 1: graphic may be improved, contents may be enriched and more detailed

line 142 repeats in part line 129

Table 2: COCAA-COLA study: check editing (different character column 9)

Author Response

Reviewer #1; Comment#1

Table 1: graphic may be imporved

Reviewer #1; Answer#1

We have altered the graphic design accordingly. Please find the revised Table 1attached below

Therapeutic Strategy

Agent/Approach tested

Immunization

active immunity : vaccines

passive immunity: plasma, immunoglobulins

Antiviral Agents

fusion inhibitors

RNA polymerase inhibitors

protease inhibitors

endosome acidification inhibitors

Supportive  Care

ECMO

artificial liver system (ALS)

cytokine filters

thromboprophylaxis

Immunomodulation

Interferons

Steroids

Colchicine

Macrolides

JAK inibitors

BTK inhibitors

IL-6 inhibitors

IL-1 inhibitors

Anti-GM-CSF Abs

Thalidomide

Cell therapies

Reviewer #1; Comment#2

Fig. 1: graphic may be improved, contents may be enriched and more detailed

Reviewer #1; Answer#2

Please find below the modified Figure 1 and a new improved Figure legend, discussing the main hypothesis and including abbreviations.

Figure 1 Legend: Contributory mechanisms of hypercoagulability in COVID 19 infection are autoimmunity and anti-phospholipid antibodies, cytokine storm-mediated thrombin overproduction, increased tissue factor and plasminogen activator inhibitor-2 levels due to ARDS-related tissue damage and ACE2 receptor-mediated endothelial dysfunction. APA anti-phospholipid antibodies, IL1RA interleukin 1 receptor agonist, IL-2R interleukin 2 receptor, TNF tissue necrosis factor, ACE angiotensin converting enzyme, ARDS acute respiratory distress syndrome TF tissue factor, PAI-2 plasminogen activator inhibitor-2.

Reviewer #1; Comment#3

line 142 repeats in part line 129

Reviewer #1; Answer#3

Line 129: ..the first clinical application of colchicine dates back to 16th century BC, when Egyptians used the herb Colchicum autumnale..

Line 142: Colchicine, a lipophilic alkaloid derived from the plant Colchicum autumnale, is metabolized in the liver and primarily excreted via the gastrointestinal tract (80%)

Line 142 was changed to’’Colchicine is a lipophilic alkaloid and it is metabolized in the liver and primarily excreted via the gastrointestinal tract (80%)’’

Reviewer #1; Comment#4

Table 2: COCAA-COLA study: check editing (different character column 9)

Reviewer #1; Answer#4

We have edited column 9 accordingly.

Reviewer 2 Report

In this review, Vrachatis and colleagues summarize the current evidence on the role of colchicine in COVID-19 prevention and treatment. The paper is well written. I have some comments for the authors which should be addressed.

1) Figure 1 would benefit from a more detailed legend, briefly summarizing the main points that the figure illustrates. Also, abbreviations should be explained in the figure legend (e.g. ARDS, IL, APA, TF, etc.) for the sake of the reader. Otherwise, a very nice figure.

2) Only one database (PubMed) was searched. This is a considerable limitation of the study and should be addressed.

3) Figure 3: The numbers do not add up? 172 – 148 does not equal 52. I assume that the 148 excluded also covers studies identified by the snowball method, as described? If so, this should be clarified. Also, the 25 studies excluded during full-text reading (52 – 27) should be shown in the figure, along with reasons for exclusion, similar to the 148 studies excluded during abstract screening.

4) Table 2: While the contents of the table support the text nicely, it is very difficult to read due to the formatting. I suggest reformatting, making the “Author, year, country” and “Outcome” columns much narrower to allow more space for the “Design”, “Dosage” and “Findings” columns.

5) Page 6: You cite several recent meta-analyses on the benefit of colchicine in COVID-19 (Vrachatis et al, Salah et al, Hariyanto et al, Chiu et al, Golpour et al, Elshafei et al, Nawangsih et al, Lien et al). Seemingly, they all indicate that colchicine is beneficial. Still, you conclude that “ongoing research will further clarify the role of colchicine in the COVID-19 pandemic.” Could you discuss the meta-analyses in a little more detail, including their limitations, do the include RCT’s only or also observational studies, etc? They could perhaps be included in Table 2, or shown in a separate table.

In line with this, I think that your conclusion could be a little more firm. If there is not sufficient evidence, or if the evidence is not of sufficient quality to conclude on the role of colchicine n COVID-19 disease, this should be stated. 

6) Did you consider performing an updated meta-analysis yourselves? Perhaps with data from the RECOVERY study, if these are available.

Minor: 

1) Heading section 2 (page 1, line 42), “SARS-COV-2 Clinical and Pathophysiological Aspects”: Consider using the term “COVID-19” instead of “SARS-COV-2”, as you are describing clinical and pathophysiological aspects of the disease.

2) You change between “SARS-COV-2” and “SARS-CoV-2”, please be consistent.

3) I suggest formatting of Table 1 more in line with the general table layout of the journal. Also, especially in the cells “Antiviral agents” and “Immunomodulation”, you could put the different items on separate lines to increase legibility.

Author Response

Reviewer#2; Comment#1

Figure 1 would benefit from a more detailed legend, briefly summarizing the main points that the figure illustrates. Also, abbreviations should be explained in the figure legend (e.g. ARDS, IL, APA, TF, etc.) for the sake of the reader. Otherwise, a very nice figure.

Reviewer#2; Answer#1

Please find below the modified Figure 1 and a new improved Figure legend, discussing the main hypothesis and including abbreviations.

Figure 1 Legend: Contributory mechanisms of hypercoagulability in COVID 19 infection are autoimmunity and anti-phospholipid antibodies, cytokine storm-mediated thrombin overproduction, increased tissue factor and plasminogen activator inhibitor-2 levels due to ARDS-related tissue damage and ACE2 receptor-mediated endothelial dysfunction. APA anti-phospholipid antibodies, IL1RA interleukin 1 receptor agonist, IL-2R interleukin 2 receptor, TNF tissue necrosis factor, ACE angiotensin converting enzyme, ARDS acute respiratory distress syndrome, TF tissue factor, PAI-2 plasminogen activator inhibitor-2.

Reviewer#2; Comment#2

 Only one database (PubMed) was searched. This is a considerable limitation of the study and should be addressed.

Reviewer#2; Answer#2

This comment was incorporated ‘’ Finally, a limitation of our study is that we opted for data published in MEDLINE instead of unpublished papers or studies published in other databases.’’

Reviewer#2; Comment#3

Figure 3: The numbers do not add up? 172 – 148 does not equal 52. I assume that the 148 excluded also covers studies identified by the snowball method, as described? If so, this should be clarified. Also, the 25 studies excluded during full-text reading (52 – 27) should be shown in the figure, along with reasons for exclusion, similar to the 148 studies excluded during abstract screening.

Reviewer#2; Answer#3

Number 52 refers to the number of full-text studies under initial evaluation and 27 is the number of the finally selected/eligible studies for discussion in our text. We have changed Figure 3 to depict this clearly.

Reviewer#2; Comment#4

 Table 2: While the contents of the table support the text nicely, it is very difficult to read due to the formatting. I suggest reformatting, making the “Author, year, country” and “Outcome” columns much narrower to allow more space for the “Design”, “Dosage” and “Findings” columns.

Reviewer#2; Answer#4

We have made changes accordingly.  Apparently it is an issue that an editorial assistant can handle since, the table’s columns are not distorted in the uploaded word file. I have tried to insert the dataset in the journal’s templates, yet the columns are always distorted after automatic PDF formatting. We can share the table in a world file or our PDF file, which is not distorted, if needed so.

Author           Country      Year

Design

N

Age

♀

(%)

BMI (kg/m2)

DM (%)

Colchicine Dosage

SoC

Findings

Outcome

Randomized Controlled Trials

Deftereos SG[78]

Greece

April 2020

prospective, open-label, RCT

105

64

41.9

27.5

10.5

L:2 mg

M: 0.5 mg BD

up to  21 days

HCQ ± AZM           ± LPV/r                    ± tocilizumab

WHO-OSCD: 1.8% vs 14%

OR 0.11; 95% CI: 0.01-0.96; P: 0.02

Positive

Tardif JC et al[81]

Multinational

March 2020 - January 2021

double-blinded RCT

non-hospitalized patients

4488

54

53.9

30

19.9

0.5 mg BD 3d

0.5mg OD 27d

HCQ

OACs

Anti-platelets

Composite of death or hospitalization

4.7% vs 5.8%

OR:0.79; 95% CI:0.61-1.03; P:0.081

Secondary endpoint, Mechanical ventilation

OR:0.53; 95% CI:0.25-1.09

secondary analysis PCR-confirmed COVID-19

Composite of death or hospitalization

4.6% vs 6%

OR:0.75; 95% CI:0.57 -0.99; P:0.042

 Any hospital admission (not pre-specified)

OR :0.76; 95% CI:0.58–0.99; p:0.04

Neutral

Lopes M. et al[94]

Brazil

April -August 2020

double blinded RCT

72

55

54

31.6

39

0.5mg q8hrs 5 days

0.5mg BD 5 days

AZM 500mg OD  up to 7 days

HCQ 400mg BD daily for 2 days, then 400mg OD  up to 8 days

Need for supplemental oxygen

4.0 days vs 6.5 days; p<0.001

Time of hospitalization

7.0 days vs 9.0 days; p:0.003

Positive

Mareev VY et al[95]

(COLORIT study)

Russia

prospective comparative RCT

43

61

30

30.4

11.6

1mg/day for 3 days

0.5mg/day for 14 days

NA

Primary endpoint

SHOCS-COVID score change day 12

8 to 2 vs 7 to 7 (p:0.002)

Secondary outcomes

Hospital stay: 13 vs 17.5; p:0.079

Any oxygen support: decreased from 50% to 9.5%; p:0.011

Deaths: 0 vs 2; p:0.467

Positive

Horby PW et al[82]

Pre-print

RECOVERY

NCT04381936 

Multinational

November 2020 -March 2021  

open-label, RCT

11340

63.4

30.5

NA

25.5

L: 1mg + 0.5mg 12h later

M: 0.5mg BD up to 10 days

corticosteroids

remdesivir

tocilizumab

convalescent plasma

baricitinib

aspirin

Primary outcome

28-day all-cause mortality

RR:1.01;95% CI: 0.93 -1.0; p:0.77

Secondary outcomes

time to discharge 10 vs 10 d

invasive mechanical ventilation 11 vs 11%

Neutral

Observational studies

Brunetti L et al [96]

USA

March-May 2020

propensity matched retrospective observational cohort

66

62.9

34.8

30.7

21.2

L: 1.2mg

M: 0.6mg BD

AZM, HCQ                 ±              remdesivir or tocilizumab

28-days mortality

 OR:0.20; 95% Cl:0.05–0.80; p : 0.023

WHO-OSCI  days 14 and 28

57.6% vs 51.5%; p : 0.621

Not requiring supplemental oxygen on days 14 and 28

54.5% vs 54.5%, p : 1.0

Hospital discharge by day 28

OR:5.0; 95% CI:1.25–20.1; p : 0.023

Positive

Scarsi M et al[97]

Italy

March -April 2020

retrospective, case-control

observational study

262

70

36.5

NA

NA

M: 1 mg/day

HCQ                             ±

dexamethasone  ±

LPV/r

21 days survival rate:

84.2% vs 63.6%, p:0.001

adj HR:0.151; 95% CI: 0.062- 0.368); p<0.0001

Positive

Sandhu T et al[98]

USA

March-May 2020

prospective comparative cohort study

112

67

43

27.5

42

0.6mg BD 3 days

0.6mg OD for 9 days

HCQ, steroids, oseltmamivir

Excluded if on: lamivudine dolutegravir, tocilizumab  convalescent plasma

Mortality: 49.1% vs 72.9%; p: 0.002

Intubations: 52.8% vs 73.6%; p: 0.006

Discharge rate: 50.9% vs 27.1%; p :0.002

Positive

Kevorkian JP et al [99]

(COCAA-COLA study)

France

January-November 2020

observational cohort study

68

66

22

27

44

Prednisone 1mg/kg/day

Furosemide

80mg/day

Salicylate

75mg/day

Colchicine

L: 1.5 mg

M: 0.5 mg q8hrs

Rivaroxaban or Apixaban

dexamethasone

(6 mg OD for up to 10 days)

LMWH

Primary composite endpoint

OR : 0.097; 95% CI:0.001–0.48;P : 0.0009

  • requirement of HFOT: 13% vs  4%; p:0.38
  • non-invasive mechanical ventilation: 13% vs 0%; p:0.07
  • invasive mechanical ventilation: 15% vs 4%; p: 0.21
  • 28-day mortality: 5% vs 0%; p:0.5

Positive

Manenti L et al [100]

Italy

February-April  2020

retrospective cohort study

age & sex matched

141

61.5

29

27.5

17

M: 1mg/day

up to 21 days

ABX, antivirals, HCQ, i.v steroids, tocilizumab

21 days mortality

adj HR:0.24; 95%CI:0.09-0.67; P : 0.006

WHO-OSCI :

adj relative improvement rate 1.80

95% CI:1.00 -3.22; p : 0.048

Positive

García-Posada M et al[101]

Colombia

May- August 2020

descriptive

observational study

209

60

39

NA

25.3

L:2 mg

M: 0.5 mg BD

up to  20 days

varying combinations of

ABX corticosteroids

LMWH

or tocilizumab

All-cause mortality:

(combination of ABX,LMWH, colchicine, corticosteroids)

OR: 0.26; 95% CI:0.08−0.71; p<0.05

positive

Abbreviations; N: number of study participants, ♀ female symbol, DM: diabetes mellitus, SoC: standard of care, L: loading dose, M: maintenance dose, ABX: antibiotics, BD: twice daily, OD: once daily, RCT: randomized clinical trial, WHO-OSCI: ordinal scale clinical improvement, WHO-OSCD: ordinal scale clinical deterioration, LPV/r: Lopinavir /ritonavir, AZM: azithromycin, HCQ: hydroxychloroquine, OACs: oral anticoagulants, LMWH: low molecular weight heparin, HFOT: high flow oxygen therapy, SHOCS-COVID score: assessment of the patient’s clinical condition, computed tomography (CT) score of the lung tissue damage, CRP changes, D-dimer changes

Reviewer#2; Comment#5

Page 6: You cite several recent meta-analyses on the benefit of colchicine in COVID-19 (Vrachatis et al, Salah et al, Hariyanto et al, Chiu et al, Golpour et al, Elshafei et al, Nawangsih et al, Lien et al). Seemingly, they all indicate that colchicine is beneficial. Still, you conclude that “ongoing research will further clarify the role of colchicine in the COVID-19 pandemic.” Could you discuss the meta-analyses in a little more detail, including their limitations, do the include RCT’s only or also observational studies, etc? They could perhaps be included in Table 2, or shown in a separate table. In line with this, I think that your conclusion could be a little more firm. If there is not sufficient evidence, or if the evidence is not of sufficient quality to conclude on the role of colchicine n COVID-19 disease, this should be stated.

Reviewer#2; Answer#5

The following comment was added: ‘’ Of note, Nawangsih et al. demonstrated that colchicine’s survival benefit is more pronounced in younger adults yet, remains statistically important irrespective of diabetes and coronary artery disease history[91]. The incorporation of RECOVERY trial findings in a meta-analysis conducted by Lien et al did not alter the afore-mentioned survival benefit among colchicine treated patients [92] (see also Table 3)’’

Table 3 Legend: 8 meta-analyses reporting effect of colchicine administration in COVID-19 mortality. (Abbreviations: OR odds ratio, RR relative risk, CI confidence interval, N/R not reported, N population, RCTs randomized clinical trials)

Meta-analysis

Journal

N

Studies Included

Mortality effect

Vrachatis et al 2021

Hell J Cardiol

881

6 studies

3 cohorts

2 RCTs

1 case-control

OR 0.35

(95%CI: 0.24-0.52; p<0.05)

Salah et al 2021

Am J Cardiol

5259

8 studies

4 cohorts

3 RCTs

1 case-control

RR 0.62

(95% CI: 0.48-0.81; P=0.0005)

Hariyanto  et al 2021

Clin Exp Pharmacol Physiol

5778

8 studies

4 cohorts

3 RCTs

1 case-control

OR 0.43

(95% CI: 0.32–0.58; p=N/R)

Chiu et al 2021

medRxiv (pre-print)

5033

6 studies

3 cohorts

3 RCTs

OR 0.36

(95% CI: 0.17-0.76; p= N/R)

Golpour et al 2021

Int J Immunopathol Pharmacol

5678

10 studies

5 cohorts

4 RCTs

1 case-control

RR 0.365

(95% CI: 0.555–0.748; p<0.05)

Elshafei et al 2021

Eur J Clin Invest

5522

9 studies

4 cohorts

4 RCTs

1 case-control

OR 0.35

(95% CI: 0.25-0.48; p=N/R)

Nawangsih et al 2021

Int Immunopharmacol

5530

8 studies

5 cohorts

3 RCTs

OR 0.47

(95% CI: 0.31-0.72; p = 0.001)

Lien et al 2021

Life (Basel)

17205

11 studies

7 cohorts

4 RCTs

OR 0.57

(95% CI: 0.38-0.87; p < 0.01)

The conclusion has been altered to ‘’ Currently available data have shown colchicine’s potential in SARS-COV2 infected patients. Further research will clarify its role in the ongoing COVID-19 pandemic.’’

Reviewer#2; Comment#6

Did you consider performing an updated meta-analysis yourselves? Perhaps with data from the RECOVERY study, if these are available.

Reviewer#2; Answer#6

The first meta-analysis in the field was published by our team (Vrachatis et al). Conducting a new meta-analysis is beyond authors’ scope. We endeavored to present the majority of the currently available data published in MEDLINE. We have also removed the comment ‘’ Of note, all of these meta-analyses did not include the results of the RECOVERY trial.’’  (Line 293) since, the most recent one (ref 92) included RECOVERY.

Reviewer#2; Comment#7

Heading section 2 (page 1, line 42), “SARS-COV-2 Clinical and Pathophysiological Aspects”: Consider using the term “COVID-19” instead of “SARS-COV-2”, as you are describing clinical and pathophysiological aspects of the disease

Reviewer#2; Answer#7

We have adopted the suggested modification.

Reviewer#2; Comment#8

You change between “SARS-COV-2” and “SARS-CoV-2”, please be consistent.

Reviewer#2; Answer#8

We have changed applicable entries into SARS-COV-2

Reviewer#2; Comment#9

I suggest formatting of Table 1 more in line with the general table layout of the journal. Also, especially in the cells “Antiviral agents” and “Immunomodulation”, you could put the different items on separate lines to increase legibility.

Reviewer#2; Answer#9

Please find below the modified Table 1

Therapeutic Strategy

Agent/Approach tested

Immunization

active immunity : vaccines

passive immunity: plasma, immunoglobulins

Antiviral Agents

fusion inhibitors

RNA polymerase inhibitors

protease inhibitors

endosome acidification inhibitors

Supportive  Care

ECMO

artificial liver system (ALS)

cytokine filters

thromboprophylaxis

Immunomodulation

Interferons

Steroids

Colchicine

Macrolides

JAK inibitors

BTK inhibitors

IL-6 inhibitors

IL-1 inhibitors

Anti-GM-CSF Abs

Thalidomide

Cell therapies

Reviewer 3 Report

Vrachatis et al. present a manuscript reviewing knowledge to date on the use of colchicine in COVID-19. The major critique is that this manuscript does not provide a significant contribution to the field. The majority of studies authors discuss are already discussed elsewhere. How do you think your manuscript will benefit the field the COVID-19 research community?  

Also: 

  1. The title of the manuscript is not supported by the conclusion.
  2. There is at least one more very important mechanism of hypercoagulability in COVID-19 that is under investigation and that is an activation of platelet FcYRIIa by SARS-CoV-2-IGG complexes. This would be of direct interest of current manuscript that is trying to describe the role of immunomodulation in COVID-19. 
  3. Line 99 - citation 24 is mainly focused around the role of C5a as netrophiloattractant, and not C3a. It is known that C3a is not as potent as C5a in this role. 
  4. Line 158 - isn't it that inhibition of tubulin polymerization is responsible for colchicine's effect in chemotaxis? 
  5. Sentence starting in line 88 - needs citations.
  6. Multiple citations are required for each of the statements in lines 103-107.
  7. Line 38 - COVID-19 is a clinical entity and therefore it should be SARS-CoV-2 in parenthesis rather than COVID-19.
  8. Sentence in line 45 needs citation.
  9. Not sure "molecular mimicry" terminology is suitable in the sentence in line 52. 
  10. Line 70 - nasal swabs as well as sputum are also used. 
  11. Citation 27 is a case report and therefore not suitable as a reference in line 112.

Author Response

Reviewer#3; Comment#1

Vrachatis et al. present a manuscript reviewing knowledge to date on the use of colchicine in COVID-19. The major critique is that this manuscript does not provide a significant contribution to the field. The majority of studies authors discuss are already discussed elsewhere. How do you think your manuscript will benefit the field the COVID-19 research community?  

Reviewer#3; Answer#1

Our manuscript highlights the multifaceted hypothesis of colchicine’s pleiotropic effects on COVID-19 infection. Our paper boasts a thorough presentation of pre-clinical and clinical data of colchicine in parallel with educative tables and figures summarizing current knowledge. To the best of our knowledge no other review paper combines the above mentioned features.

Reviewer#3; Comment#2

The title of the manuscript is not supported by the conclusion.

Reviewer#3; Answer#2

The conclusion has been altered to ‘’ Currently available data have shown colchicine’s potential in SARS-COV2 infected patients. Further research will clarify its role in the ongoing COVID-19 pandemic.’’

Reviewer#3; Comment#3

There is at least one more very important mechanism of hypercoagulability in COVID-19 that is under investigation and that is an activation of platelet FcYRIIa by SARS-CoV-2-IGG complexes. This would be of direct interest of current manuscript that is trying to describe the role of immunomodulation in COVID-19.

Reviewer#3; Answer#3

The following comment was added: ‘’Immune complexes containing antibodies against SARS-COV2 spike protein are also implicated in COVID-19 hypercoagulability through enhanced of FcγRIIA signaling [39,40].’’

[39] Bye AP, Hoepel W, Mitchell JL, Jégouic S, Loureiro S, Sage T, et al. Aberrant glycosylation of anti-SARS-CoV-2 IgG is a pro-thrombotic stimulus for platelets. BioRxiv 2021:2021.03.26.437014. https://doi.org/10.1101/2021.03.26.437014.

[40] Nazy I, Jevtic SD, Moore JC, Huynh A, Smith JW, Kelton JG, et al. Platelet-activating immune complexes identified in critically ill COVID-19 patients suspected of heparin-induced thrombocytopenia. J Thromb Haemost 2021;19:1342–7. https://doi.org/10.1111/jth.15283.

Reviewer#3; Comment#4

Line 99 - citation 24 is mainly focused around the role of C5a as netrophiloattractant, and not C3a. It is known that C3a is not as potent as C5a in this role. 

Reviewer#3; Answer#4

Line 99 was altered to: ‘’Complement component C5a, acting as chemoattractant of neutrophils, potentiates the development of a deleterious hypercytokinaemia [24]’’

Reviewer#3; Comment#5

Line 158 - isn't it that inhibition of tubulin polymerization is responsible for colchicine's effect in chemotaxis?

Reviewer#3; Answer#5

Line 158 was altered to: ‘’Colchicine seems to possess pleiotropic mechanisms of action..’’

Reviewer#3; Comment#6

Sentence starting in line 88 - needs citations

Reviewer#3; Answer#6

A citation was added at the end of this sentence (ref 20)

Reviewer#3; Comment#7

Multiple citations are required for each of the statements in lines 103-107.

Reviewer#3; Answer#7

The following references were added and the text was modified accordingly (Please see marked changes)

[26] Huang C-F, Hsieh S-M, Pan S-C, Huang Y-S, Chang S-C. Dose-Related Aberrant Inhibition of Intracellular Perforin Expression by S1 Subunit of Spike Glycoprotein That Contains Receptor-Binding Domain from SARS-CoV-2. Microorganisms 2021;9:1303. https://doi.org/10.3390/microorganisms9061303.

[27] Jenner AL, Aogo RA, Alfonso S, Crowe V, Deng X, Smith AP, et al. COVID-19 virtual patient cohort suggests immune mechanisms driving disease outcomes. PLOS Pathog 2021;17:e1009753. https://doi.org/10.1371/journal.ppat.1009753.

[28] Severa M, Diotti RA, Etna MP, Rizzo F, Fiore S, Ricci D, et al. Differential plasmacytoid dendritic cell phenotype and type I Interferon response in asymptomatic and severe COVID-19 infection. PLOS Pathog 2021;17:e1009878. https://doi.org/10.1371/journal.ppat.1009878.

[29] Barnes BJ, Adrover JM, Baxter-Stoltzfus A, Borczuk A, Cools-Lartigue J, Crawford JM, et al. Targeting potential drivers of COVID-19: Neutrophil extracellular traps. J Exp Med 2020;217. https://doi.org/10.1084/jem.20200652.

[30] Obermayer A, Jakob L-M, Haslbauer JD, Matter MS, Tzankov A, Stoiber W. Neutrophil Extracellular Traps in Fatal COVID-19-Associated Lung Injury. Dis Markers 2021;2021:1–10. https://doi.org/10.1155/2021/5566826.

[31] Ferreira AC, Soares VC, de Azevedo-Quintanilha IG, Dias S da SG, Fintelman-Rodrigues N, Sacramento CQ, et al. SARS-CoV-2 engages inflammasome and pyroptosis in human primary monocytes. Cell Death Discov 2021;7:43. https://doi.org/10.1038/s41420-021-00428-w.

Reviewer#3; Comment#8

Line 38 - COVID-19 is a clinical entity and therefore it should be SARS-CoV-2 in parenthesis rather than COVID-19.

Reviewer#3; Answer#8

The change has been made. Thank you for noticing it.

Reviewer#3; Comment#9

Sentence in line 45 needs citation.

Reviewer#3; Answer#9

Reference 1 is applicable. We have made the change.

Reviewer#3; Comment#10

Not sure "molecular mimicry" terminology is suitable in the sentence in line 52. 

Reviewer#3; Answer#10

We deleted the term molecular mimicry.

Reviewer#3; Comment#11

Line 70 - nasal swabs as well as sputum are also used. 

Reviewer#3; Answer#11

This term was added.

Reviewer#3; Comment#12

Citation 27 is a case report and therefore not suitable as a reference in line 112

Reviewer#3; Answer#12

The following citations have been used to replace it.

[33] Foret T, Dufrost V, Salomon Du Mont L, Costa P, Lefevre B, Lacolley P, et al. Systematic Review of Antiphospholipid Antibodies in COVID-19 Patients: Culprits or Bystanders? Curr Rheumatol Rep 2021;23:65. https://doi.org/10.1007/s11926-021-01029-3.

[34] Hollerbach A, Müller-Calleja N, Pedrosa D, Canisius A, Sprinzl MF, Falter T, et al. Pathogenic lipid-binding antiphospholipid antibodies are associated with severity of COVID-19. J Thromb Haemost 2021;19:2335–47. https://doi.org/10.1111/jth.15455.

Round 2

Reviewer 3 Report

Authors addressed the comments, however there is still a limited contribution to the field.